# Serum Anti-Aging Protein α-Klotho Mediates the Association between Diet Quality and Kidney Function

**DOI:** 10.3390/nu15122744

**Published:** 2023-06-14

**Authors:** Qingqing Cai, Shixian Hu, Cancan Qi, Jiawei Yin, Shulan Xu, Fan Fan Hou, An Li

**Affiliations:** 1Division of Nephrology, Nanfang Hospital, Southern Medical University, National Clinical Research Center for Kidney Disease, State Key Laboratory of Organ Failure Research, Guangdong Provincial Institute of Nephrology, Guangdong Provincial Key Laboratory of Renal Failure Research, Guangzhou 510515, China; qqcai@smu.edu.cn; 2Institute of Precision Medicine, The First Affiliated Hospital, Sun Yat-Sen University, Guangzhou 510080, China; hushx9@mail.sysu.edu.cn; 3Department of Gastroenterology and Hepatology, University Medical Center Groningen (UMCG), University of Groningen, 9712 Groningen, The Netherlands; 4Microbiome Medicine Center, Division of Laboratory Medicine, Zhujiang Hospital, Southern Medical University, Guangzhou 510282, China; cc_qi@hotmail.com; 5Department of Nutrition and Food Hygiene, Hubei Key Laboratory of Food Nutrition and Safety & Ministry of Education Key Laboratory of Environment and Health, School of Public Health, Tongji Medical College, Huazhong University of Science and Technology, Wuhan 430074, China; 2020509033@hust.edu.cn; 6Stomatological Hospital, School of Stomatology, Southern Medical University, Guangzhou 510280, China; xushulan_672588@smu.edu.cn

**Keywords:** α-Klotho, Healthy Eating Index 2015, kidney function, estimated glomerular filtration rate, aging

## Abstract

Adherence to healthy dietary patterns is associated with a reduced risk of kidney dysfunction. Nevertheless, the age-related mechanisms that underpin the relationship between diet and kidney function remain undetermined. This study aimed to investigate the mediating role of serum α-Klotho, an anti-aging protein, in the link between a healthy diet and kidney function. A cross-sectional study was conducted on a cohort of 12,817 individuals aged between 40 and 79 years who participated in the National Health and Nutrition Examination Survey (NHANES) from 2007 to 2016. For each participant, the Healthy Eating Index 2015 (HEI-2015) score was calculated as a measure of a healthy dietary pattern. Creatinine-based estimated glomerular filtration rate (eGFR) was used to assess kidney function. Multivariable regression models were used to analyze the association between the standardized HEI-2015 score and eGFR after adjusting for potential confounders. Causal mediation analysis was performed to assess whether serum α-Klotho influenced this association. The mean (±SD) eGFR of all individuals was 86.8 ± 19.8 mL/min per 1.73 m^2^. A high standardized HEI-2015 score was associated with a high eGFR (β [95% CI], 0.94 [0.64–1.23]; *p* < 0.001). The mediation analysis revealed that serum α-Klotho accounted for 5.6–10.5% of the association of standardized overall HEI-2015 score, total fruits, whole fruits, greens and beans, and whole grain with eGFR in the NHANES. According to the results from the subgroup analysis, serum α-Klotho exerted a mediating effect in the participants aged 60–79 years and in males. A healthy diet may promote kidney function by up-regulating serum anti-aging α-Klotho. This novel pathway suggests important implications for dietary recommendations and kidney health.

## 1. Introduction

Globally, chronic kidney disease (CKD) affects 8% to 16% of the population, making it a major public health issue [1,2]. CKD is a progressive disorder that can lead to kidney failure, cardiovascular complications, and premature mortality [3]. A healthy lifestyle that includes a healthy diet may delay the aging process and reduce the risk of various aging-related chronic diseases, including CKD [4,5]. Studies have consistently demonstrated a relationship between adherence to a healthy diet, such as the Mediterranean diet or those described by the Healthy Eating Index 2015 (HEI-2015) and the Dietary Approaches to Stop Hypertension (DASH), and a reduced risk of CKD along with an improved estimated glomerular filtration rate (eGFR) [6,7,8]. Thus, adopting a healthy diet as early as possible is crucial to prolonging kidney function. However, the age-related mechanisms underlying the link between diet quality and kidney function are not yet fully understood.

In recent years, there has been a growing interest in exploring the role of serum α-Klotho, an anti-aging protein, in relation to kidney function. α-Klotho can be expressed in proximal and distal renal tubules, and its transmembrane form has been identified as the co-receptor of fibroblast growth factor 23 (FGF23), which plays a crucial role in regulating phosphate excretion [9,10]. Secretases cleave the extracellular domain of α-Klotho, leading to the release of the soluble form of α-Klotho into the systemic circulation [11]. Serum α-Klotho functions as an endocrine factor and is implicated in numerous physiological processes, including mineral metabolism, insulin signaling, and modulation of oxidative stress [12]. Animal studies suggest that α-Klotho has a protective effect against kidney injury and fibrosis [13,14,15]. Epidemiological studies have demonstrated a positive correlation between serum α-Klotho levels and eGFR among adults aged 40–79 years, as well as an inverse association between serum α-Klotho levels and the prevalence of CKD [16]. Moreover, a positive association has been established between the serum α-Klotho level and adherence to a healthy diet, as indicated by HEI-2015 scores [17]. Nevertheless, the role of serum α-Klotho in the relationship between diet quality and kidney function has not been fully elucidated.

This study aimed to investigate whether serum α-Klotho mediates the relationship between a healthy dietary pattern and kidney function in a large cohort of middle-to-older aged individuals from the National Health and Nutrition Examination Survey (NHANES) study conducted between 2007 and 2016.

## 2. Materials and Methods

### 2.1. Data Source and Study Participants

NHANES comprises a sequence of surveys conducted on a nationally representative sample of non-institutionalized civilians in the United States aimed at evaluating their health and nutrition status [18]. We combined five survey cycles of the NHANES (2007–2016) to conduct a cross-sectional study. The sampling frame of NHANES is based on a complex, stratified, and multi-stage probability sample design. The NHANES data were gathered via in-home interviews, dietary interviews, medical examinations, and laboratory tests. The National Center for Health Statistics Ethics Review Board granted approval for all the NHANES protocols, and each participant provided written informed consent [19]. The present study adheres to the STROBE guidelines for reporting the findings [20]. The initial cohort included 50,588 participants under the age of 80 from NHANES 2007–2016. The serum α-Klotho level was assessed in participants aged 40–79 years as part of the NHANES study. Subsequently, individuals without data on serum α-Klotho (*n* = 36,824), diet quality (*n* = 804), or kidney function (*n* = 143) were sequentially excluded. The final analytical cohort consisted of 12,817 participants aged between 40 and 79 years old (Figure 1).

### 2.2. Assessment of Diet Quality

The HEI-2015 was employed to ascertain the quality of the overall diet. The dietary evaluation in NHANES was predicated on 24 h dietary recall interviews (24 h) conducted at mobile examination centers. For this analysis, the first 24 h (if two 24 h were available) and the Food Patterns Equivalents were selected to estimate diet-quality scores. The HEI-2015 is a modern instrument utilized to assess conformity with the 2015–2020 Dietary Guidelines for Americans [21,22], comprising 13 dietary components, and encompassing 9 adequacy and 4 moderation components (Appendix A). High intakes of adequacy and moderation components result in high and low scores, respectively. Each component is individually scored and subsequently summed to obtain the HEI-2015 score (range: 0–100). A higher score indicates a healthier diet quality [23].

### 2.3. Assessment of Kidney Function

The assessment of kidney function in this study was conducted through the utilization of two principal parameters, namely, the eGFR and urine albumin–creatinine ratio (UACR). The primary focus of this investigation was on the eGFR, which was determined by means of the 2012 Chronic Kidney Disease Epidemiology Collaboration equation (CKD-EPI) that relies on the serum creatinine level [24]. The UACR was employed as an outcome measure for sensitivity analysis, and its calculation was based on the following formula:UACR (mg/g) = urinary albumin level (mg/dL)/urinary creatinine level (g/dL). 

The levels of urine albumin and creatinine were obtained from spot urine samples.

### 2.4. Quantitation of Serum α-Klotho

Serum levels of α-Klotho were measured given that this protein has been proposed to suppress aging [11]. The serum samples were obtained by centrifuging whole blood from the participants and then stored at −80 °C. The α-Klotho concentrations in the pristine serum specimens were measured using a commercially available enzyme-linked immunosorbent assay kit (IBL International, Gunma, Japan) [25]. Each serum sample was analyzed in duplicate, and the mean of the two measurements was used as the final α-Klotho concentration. The analysis outcomes were forwarded to the laboratory of the Oracle Management System for assessment. In the event of a discrepancy greater than 10% between duplicate values, the measurements were re-executed. Furthermore, if the value of the quality-control sample exceeded two standard deviations of the established value, the whole plate was considered invalid and subsequently repeated. The assay had a sensitivity of 6 pg/mL, and the reference range for α-Klotho concentration was 285.8–1638.6 pg/mL.

### 2.5. Identification of the Covariates

The sociodemographic information collected comprised age, sex (male or female), race/ethnicity (non-Hispanic White, non-Hispanic Black, Hispanic, or others), education level (college or higher, high school, or below high school), and annual household income (greater than USD 75,000, USD 20,000–75,000, or less than USD 20,000). The age variable was stratified into two groups: middle-aged (40–59 years) or older adults (60 years and above). The classification of body weight was based on the body mass index (BMI) and was divided into three categories: normal (BMI < 25.0 kg/m^2^), overweight (BMI, 25.0–29.9 kg/m^2^), and obesity (BMI ≥ 30 kg/m^2^). The healthy-lifestyle variables that were considered included smoking status (current smoker, former smoker, or never smoked), physical activity level (vigorous, moderate, or low), and alcohol consumption (heavy, moderate, or never). Energy intake (kcal/day) was analyzed based on self-reported data. Self-reported hypertension, diabetes, and cardiovascular disease were regarded as potential confounding variables.

### 2.6. Statistical Analysis

Categorical variables were assessed through the calculation of counts (percentages), while parametric continuous variables were evaluated through the determination of means (±SD), and non-parametric continuous variables were analyzed through the calculation of medians (with IQRs).

First, we conducted multivariable linear regression analyses to assess the association of the HEI-2015 score with eGFR. Our analyses involved three models: Model 1, which was adjusted for age, sex, and race; Model 2, which was additionally adjusted for education, income, total energy intake, alcohol intake, smoking, and physical activity; and Model 3, which was further adjusted for BMI, self-reported hypertension, self-reported diabetes, and self-reported cardiovascular disease.

Second, we explored the potential mediating roles of the aging process in the association between diet quality and kidney function. The causal mediation analysis was used for continuous exposure, mediator, and outcome with linear regression models (Figure 2). A bootstrap method was used to assess the confidence intervals for pathway estimates. Specifically, the HEI-2015 score was defined as the exposure variable, serum anti-aging protein α-Klotho as the mediator, and eGFR as the outcome variable. The following effects were measured: (i) the effect of diet quality on the anti-aging protein (Figure 2, path a), (ii) the effect of the anti-aging protein on kidney function (Figure 2, path b), (iii) the total effect of diet quality on kidney function (Figure 2, path c), and (iv) the direct effect of diet quality on kidney function, with the anti-aging protein considered (Figure 2, path cʹ). The proportion mediated was calculated using the formula:$$\;(\beta_{\;\mathrm{total}\;\mathrm{effect}}-\beta_{\;\mathrm{direct}\;\mathrm{effect}})/\beta_{\;\mathrm{total}\;\mathrm{effect}}\times 100$$

The methodological details of the mediation analysis have been described elsewhere [26,27].

Third, a subgroup analysis was conducted to investigate whether age or sex has a modifying effect on the mediating role of serum α-Klotho in the association between standardized overall HEI-2015 score and eGFR. This analysis was stratified by age (40–59 vs. 60–79 years) or sex (male vs. female) in the mediation analysis. Additionally, we explored the role of serum α-Klotho in the relationship between standardized HEI-2015 scores and log-transformed UACR as sensitivity analyses.

Since the amount of missing data in the covariates was small (<5%), we used multivariate imputation by chain equations (MICE) to handle the missing data [28]. The statistical analyses were performed using R (version 4.1.2), with a two-sided *p*-value of less than 0.05 indicating statistical significance.

## 3. Results

### 3.1. Population Characteristics

Characteristics of the 12,817 participants aged from 40 to 79 years in the NHANES from 2007 to 2016 are presented in Table 1. The mean (±SD) age of all individuals was 57.7 ± 10.8 years, and 51.5% were female. The mean (±SD) eGFR of all individuals was 86.8 ± 19.8 mL/min per 1.73 m^2^. The mean (±SD) HEI-2015 score was 55.2 ± 13.4, and the mean (±SD) serum α-Klotho level was 855.4 ± 310.6 pg/mL.

### 3.2. Association between HEI-2015 Score and Kidney Function

The relationship between the standardized HEI-2015 score and kidney function is summarized in Table 2. After adjusting for demographic, lifestyle, and clinical covariates in Model 3, a positive association was observed between the standardized overall HEI-2015 score and eGFR (β [95% CI], 0.94 [0.64–1.23]; *p* < 0.001). Further analysis of individual food groups revealed that high standardized scores of total fruits, whole fruits, total vegetables, greens and beans, whole grain, seafood and plant protein, fatty acids, saturated fat, and added sugar were also significantly associated with a high eGFR. As a sensitivity analysis, we analyzed the association between the standardized HEI-2015 score and log-transformed UACR to confirm the findings. The results were generally consistent with those of the main analysis (Appendix A). High standardized overall HEI-2015 scores, as well as high intake of total fruits, whole fruits, total vegetables, greens and beans, whole grain, and seafood and plant protein, were found to be associated with low UACR levels. Conversely, a high intake of added sugar was associated with high UACR levels.

### 3.3. Mediating Role of Serum α-Klotho

A mediation analysis was conducted to assess whether serum α-Klotho plays a mediating role in the association between the HEI-2015 score and kidney function, with potential confounders adjusted. The results showed that the overall HEI-2015 score had a significant indirect effect on eGFR through serum α-Klotho (β [95% CI], 0.05 [0.02–0.08]; *p* < 0.001; Table 3), indicating a partial mediating effect of serum α-Klotho. The effect of the overall HEI-2015 score on eGFR remained significant after controlling for serum α-Klotho (β [95% CI], 0.88 [0.59–1.19] *p* < 0.001), suggesting the presence of both direct and indirect effects. Approximately 5.8% [95% CI, 2.6–10.0%] of the effect of the overall HEI-2015 score on eGFR was found to be mediated by serum α-Klotho. Further analysis of the individual food groups revealed that serum α-Klotho accounted for 10.5% [4.3–23.0%], 9.6% [5.1–19.0%], 5.6% [1.2–13.0%], and 8.4% [3.3–20.0%] of the association between the standardized intake of total fruits, whole fruits, greens and beans, and whole grain, respectively, and eGFR in the NHANES. However, the sensitivity analysis revealed that serum α-Klotho did not mediate the association between the HEI-2015 score and the UACR (Appendix A).

### 3.4. Subgroup Analyses

We performed subgroup analyses to examine potential effect modification by age or sex on the mediating role of serum α-Klotho in the relationship between the standardized overall HEI-2015 score and eGFR. The findings revealed that the mediating role of serum α-Klotho in the association between HEI 2015 score and eGFR was observed in males (β [95% CI], 0.09 [0.04–0.14], *p* < 0.001) and participants aged 60–79 years (β [95% CI], 0.10 [0.04–0.16], *p* < 0.001), with the proportion mediated being 9.6% [4.5–20.0%] and 7.6% [3.2–14.0%], respectively (Table 4).

## 4. Discussion

This study investigated the potential mediating role of serum α-Klotho in the association between a healthy dietary pattern (measured using the HEI-2015 score) and kidney function (based on eGFR) in a cohort of 12,817 participants aged between 40 and 79 years from the NHANES study from 2007 to 2016. After adjusting for demographic, lifestyle, and clinical covariates, our findings revealed a significant positive association between HEI-2015 and eGFR. Furthermore, the results from our mediation analyses suggest that serum α-Klotho partially mediates this association, particularly in males and older individuals. These findings provide insights into the anti-aging mechanistic pathways linking a healthy dietary pattern to improved kidney function.

The association between a healthy dietary pattern and improved kidney function has been extensively studied. In a prospective study involving 12,155 participants aged 45–64 years from the Atherosclerosis Risk in Communities (ARIC) Study, it was observed that HEI-2015 is positively associated with a reduced risk of CKD over a 24-year follow-up period [6]. Similarly, a study involving 78,346 individuals from the general Dutch population with a 3.5-year follow-up period revealed that adhering to a healthy dietary pattern, characterized by high intakes of fruits, vegetables, nuts and legumes, fish, whole grains, unsweetened dairy, coffee, and tea, as well as low intakes of red and processed meat, butter and hard margarine, and sugar-sweetened beverages, was associated with a decreased risk of CKD and a slower decline in eGFR [29]. A subgroup analysis conducted on data from the Nurse’s Health Study (NHS) has shown that a Western dietary pattern, characterized by high intakes of red and processed meat, saturated fat, and sweets, is associated with increased albuminuria levels. These findings suggest that a healthy dietary pattern, in contrast to the Western diet, has a beneficial impact on kidney function [30]. In the present study, we found that high HEI-2015 scores are positively associated with eGFR and negatively associated with albuminuria after adjustment for potential covariates. These findings are consistent with previous studies and highlight the potential benefits of adhering to a healthy dietary pattern that emphasizes the consumption of plant-based foods, lean proteins, and healthy fats, while minimizing the intake of processed foods, added sugars, and unhealthy fats. Overall, our results underscore the importance of a healthy dietary pattern for promoting kidney function. However, the mechanism underlying this association is not fully understood.

The present study revealed a novel mediating effect of serum α-Klotho on the relationship between a healthy dietary pattern and improved kidney function. A previous study has demonstrated a positive association between HEI-2015 score and serum α-Klotho among 8456 participants aged 40–79 years from the NHANES study [17]. A high-quality diet is related to lower serum levels of various inflammatory factors such as IL-6, IL-1β, and TGF-β, which down-regulate the expression of α-Klotho [31,32]. Our results support the hypothesis that a healthy dietary pattern improves kidney function by up-regulating serum α-Klotho, thereby presumably slowing the aging process. α-Klotho is an anti-aging protein primarily produced in the kidneys [7,8,9]. A previous study has shown that Phenotypic Age Acceleration, a measure of accelerated aging, partially mediates the association of unhealthy lifestyles, including an unhealthy diet, with adverse health outcomes, such as cardiovascular disease, cancer, and mortality, in populations from the UK and the US [33]. Our findings suggest an anti-aging mechanism underlying the relationship between a healthy diet and enhanced kidney function through the involvement of serum α-Klotho, which plays a crucial role in preserving kidney health by regulating calcium and phosphate metabolisms, inhibiting renal fibrosis, reducing oxidative stress, and promoting anti-inflammatory effects [7,8,9,10,11,12,13,14,15]. However, our sensitivity analysis revealed no evidence of serum α-Klotho mediating the link between the HEI-2015 score and UACR. Thus, the negative association between the HEI-2015 score and UACR cannot be explained by serum α-Klotho. Dietary factors may have different effects on UACR and eGFR. Furthermore, it is worth mentioning that the role of serum α-Klotho in mediating the relationship between a healthy diet and improved kidney function may have been discovered by chance. To confirm this anti-aging mechanism, other aging-related markers, such as Phenotypic Age Acceleration, may be utilized in future studies.

The findings of this study have significant implications for dietary recommendations and the preservation of kidney health. Our results emphasize the importance of adhering to a healthy dietary pattern to maintain kidney health and implicate the involvement of the anti-aging process in this association. Specifically, we observed the important role of the serum anti-aging protein α-Klotho in the mechanism linking a healthy dietary pattern to improved kidney function, particularly among older adults and males. Our findings suggest that consuming total fruits, whole fruits, greens and beans, and whole grains promote kidney health through an anti-aging process involving serum α-Klotho. Implementation of public health interventions to promote healthy dietary habits may have benefits in preventing kidney diseases and prolonging kidney function during the aging process. Targeted dietary recommendations for males and individuals older than 60 years are especially important. Our findings uncovered a novel pathway connecting a healthy diet to improved kidney function and suggest the possibility of customizing dietary interventions to enhance kidney function by decelerating the aging process.

This study possesses several strengths, including its large sample size and the population-based design. Additionally, it contributes a novel understanding of the mechanistic pathway linking a healthy diet to improved kidney function through the mediation of serum α-Klotho. However, several limitations should be considered when interpreting the findings. First, we observed that serum α-Klotho mediates the relationship between the HEI-2015 score and eGFR, but not between the HEI-2015 score and UACR. Consequently, the negative relationship between the HEI-2015 score and UACR cannot be attributed to serum α-Klotho. Further investigations using other aging-related markers, such as telomere shortening, phenotypic age acceleration, or epigenetic age acceleration, are warranted to elucidate the mechanism linking the HEI-2015 score to kidney function. Second, although we carefully adjusted for a broad range of confounders, residual confounding may still be present. Third, causal relationships cannot be established in this study due to its cross-sectional design. Thus, longitudinal observational studies with extended follow-up periods are needed to comprehensively understand the relationship between diet quality and kidney function and to confirm the role of serum α-Klotho in the underlying biological mechanism.

## 5. Conclusions

Our study highlights the positive correlation between adhering to a healthy diet and enhanced kidney function among middle-to-older aged adults. Furthermore, our findings imply a significant role of the serum anti-aging protein α-Klotho in the mechanism linking a healthy diet to improved kidney function. This study provides novel insights into the age-related mechanisms that connect diet quality and kidney function, which may help tailor dietary interventions to promote kidney health by decelerating the aging process, especially in males and older adults. These findings have important implications for dietary recommendations and the promotion of kidney health. However, future longitudinal and prospective studies are required to validate and reinforce these findings.

## Figures and Tables

**Figure 1 nutrients-15-02744-f001:**
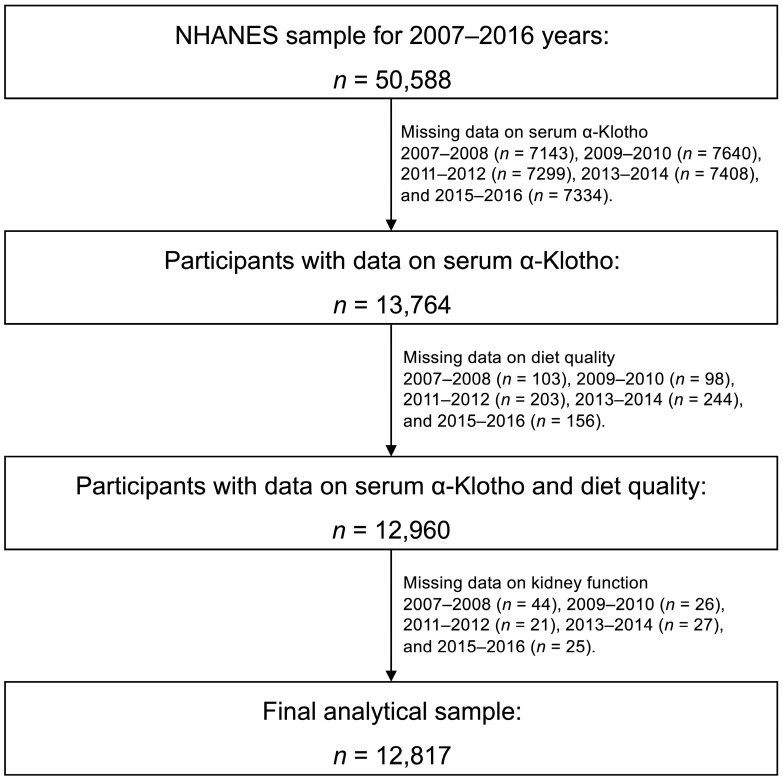
Flowchart for selecting analytical sample from the National Health and Nutrition Examination Survey (NHANES).

**Figure 2 nutrients-15-02744-f002:**
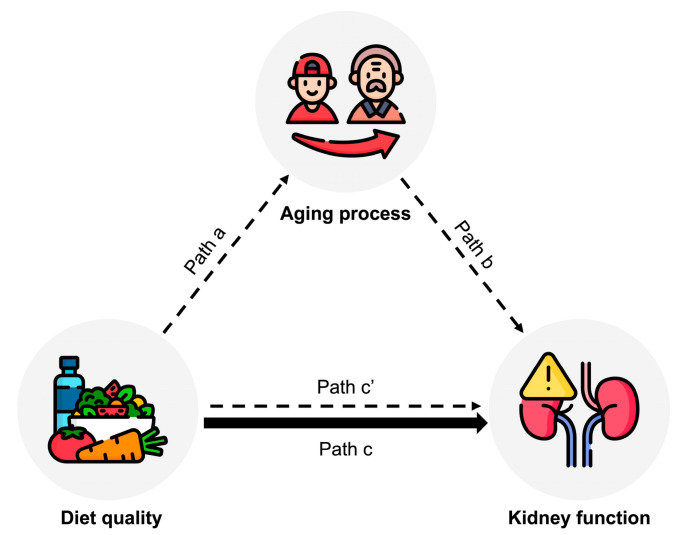
Path diagram of the mediation analysis model among 12,817 participants aged 40–79 years in the NHANES 2007–2016. Diet quality (measured by HEI-2015) was defined as an exposure; kidney function (measured by eGFR) was defined as an outcome; and aging process (measured by α-Klotho) was defined as a mediator. Path a indicates the regression coefficient for the association of HEI-2015 with α-Klotho. Path b indicates the regression coefficient for the association of α-Klotho with eGFR. Path c indicates the simple total effect of HEI-2015 with eGFR, without the adjustment for α-Klotho. Path c’ indicates the direct effect of HEI-2015 with eGFR when controlled for α-Klotho. Abbreviations: NHANES, National Health and Nutrition Examination Survey; HEI, healthy eating index; eGFR, estimated glomerular filtration rate.

**Table 1 nutrients-15-02744-t001:** Characteristics of 12,817 participants aged 40–79 years in the NHANES from 2007 to 2016.

	Whole Population
Number	12,817
HEI-2015	55.2 ± 13.4
eGFR (mL/min per 1.73 m^2^)	86.8 ± 19.8
UACR (mg/g)	7.6 (4.9–15.1)
α-Klotho (pg/mL)	855.4 ± 310.6
Age (years)	57.7 ± 10.8
Sex (Female, %)	51.5
Race (%)	
Non-Hispanic white	44.1
Non-Hispanic Black	19.7
Hispanic	27.4
Others	8.7
Body weight	
Normal weight	23.7
Overweight	34.7
Obesity	41.6
Self-reported hypertension (yes, %)	46.5
Self-reported diabetes (yes, %)	18.3
Self-reported cardiovascular disease (yes, %)	10.6
Smoking status (%)	
Current	19.5
Former	29.4
Never	51.1
Physical activity (%)	
Vigorous	13.6
Moderate	21.0
Low	65.4
Energy intake (kcal/day)	3720 ± 1597
Alcohol intake (%)	
Heavy	16.2
Moderate	35.2
Never	48.6
Education level (%)	
College or higher	50.3
High school	22.2
Less than high school	27.5
Annual household income (USD, %)	
>75,000	25.8
20,000–75,000	50.3
<20,000	23.8

Data are presented as means ± standard deviations, medians (interquartile ranges) or percentages, according to the case. Abbreviations: NHANES, National Health and Nutrition Examination Survey; HEI, healthy eating index; eGFR, estimated glomerular filtration rate; UACR, urine albumin-creatinine ratio.

**Table 2 nutrients-15-02744-t002:** Association between standardized HEI 2015 scores and eGFR among 12,817 participants aged 40–79 years in the NHANES.

	Weighted β Coefficient (95% CI) for eGFR		
	Model 1	Model 2	Model 3
Overall HEI-2015	0.86 (0.57, 1.14) *	1.11 (0.81, 1.41) *	0.94 (0.64, 1.23) *
Total fruits	0.47 (0.18, 0.76) *	0.68 (0.39, 0.98) *	0.55 (0.26, 0.85) *
Whole fruits	0.59 (0.30, 0.88) *	0.78 (0.48, 1.08) *	0.69 (0.40, 0.99) *
Total vegetables	0.47 (0.18, 0.76) *	0.60 (0.31, 0.89) *	0.59 (0.30, 0.88) *
Greens and beans	0.65 (0.36, 0.94) *	0.72 (0.43, 1.00) *	0.63 (0.34, 0.92) *
Whole grain	0.43 (0.14, 0.71) *	0.59 (0.29, 0.88) *	0.56 (0.27, 0.85) *
Refined grain	−0.06 (−0.36, 0.23)	−0.01 (−0.30, 0.29)	−0.14 (−0.43, 0.16)
Total dairy	0.14 (−0.15, 0.44)	0.21 (−0.09, 0.50)	0.17 (−0.12, 0.46)
Total protein	−0.14 (−0.42, 0.15)	−0.11 (−0.39, 0.18)	−0.02 (−0.30, 0.27)
Seafood and plant protein	0.62 (0.33, 0.90) *	0.70 (0.41, 0.99) *	0.63 (0.34, 0.92) *
Fatty acid	0.26 (−0.03, 0.54)	0.31 (0.03, 0.60) *	0.31 (0.02, 0.59) *
Saturated fat	0.40 (0.11, 0.69) *	0.44 (0.15, 0.73) *	0.31 (0.02, 0.60) *
Sodium	0.18 (−0.11, 0.46)	0.08 (−0.20, 0.37)	−0.17 (−0.46, 0.12)
Added sugar	0.49 (0.20, 0.78) *	0.64 (0.34, 0.93) *	0.76 (0.47, 1.06) *

Model 1: adjusted for age, sex and race; Model 2: model plus education, income, total energy intake, alcohol intake, smoking, and physical activity; Model 3: Model 2 plus BMI, self-reported hypertension, self-reported diabetes, and self-reported cardiovascular disease; * *p* < 0.05.

**Table 3 nutrients-15-02744-t003:** Mediating role of α-klotho in the association between standardized HEI 2015 scores and eGFR among 12,817 participants aged 40–79 years in the NHANES.

	Weighted β Coefficient (95% CI) for eGFR			
Mediator: α-Klotho	Total Effect	Direct Effect	Indirect Effect	Proportion Mediated, % (95%CI)
Overall HEI-2015	0.94 (0.64, 1.23) *	0.88 (0.59, 1.19) *	0.05 (0.02, 0.08) *	5.8 (2.6, 10.0) *
Total fruits	0.55 (0.26, 0.85) *	0.49 (0.20, 0.82) *	0.06 (0.03, 0.09) *	10.5 (4.3, 23.0) *
Whole fruits	0.69 (0.40, 0.99) *	0.62 (0.31, 0.90) *	0.07 (0.04, 0.10) *	9.6 (5.1, 19.0) *
Total vegetables	0.59 (0.30, 0.88) *	0.57 (0.27, 0.87) *	0.02 (−0.01, 0.06)	-
Greens and beans	0.63 (0.34, 0.92) *	0.60 (0.33, 0.88) *	0.04 (0.01, 0.07) *	5.6 (1.2, 13.0) *
Whole grain	0.56 (0.27, 0.85) *	0.51 (0.21, 0.82) *	0.05 (0.02, 0.08) *	8.4 (3.3, 20.0) *
Refined grain	−0.14 (−0.43, 0.16)	−0.16 (−0.45, 0.12)	0.02 (−0.01, 0.05)	-
Total dairy	0.17 (−0.12, 0.46)	0.13 (−0.15, 0.41)	0.04 (−0.01, 0.07)	-
Total protein	−0.02 (−0.30, 0.27)	−0.04 (−0.34, 0.23)	0.03 (−0.01, 0.06)	-
Seafood and plant protein	0.63 (0.34, 0.92) *	0.60 (0.31, 0.91) *	0.03 (−0.01, 0.05)	-
Fatty acid	0.31 (0.02, 0.59) *	0.29 (−0.01, 0.60)	0.02 (−0.01, 0.05)	-
Saturated fat	0.31 (0.02, 0.60) *	0.32 (0.02, 0.61) *	−0.01 (−0.04, 0.02)	-
Sodium	−0.17 (−0.46, 0.12)	−0.15 (−0.45, 0.12)	−0.02 (−0.05, 0.01)	-
Added sugar	0.76 (0.47, 1.06) *	0.79 (0.50, 1.08) *	−0.02 (−0.06, 0.01)	-

Model adjusted for age, sex, race, education, income, total energy intake, alcohol intake, smoking, physical activity, BMI, self-reported hypertension, self-reported diabetes, and self-reported cardiovascular disease; * *p* < 0.05.

**Table 4 nutrients-15-02744-t004:** Subgroup analyses of the mediating role of serum α-Klotho in the association between standardized overall HEI 2015 scores and eGFR in the NHANES.

	Weighted β Coefficient (95%CI) for eGFR			
Mediator: α-Klotho	Total Effect	Direct Effect	Indirect Effect	Proportion Mediated % (95% CI)
Age (years)				
40–59	0.59 (0.21, 0.96) *	0.56 (0.18, 0.93) *	0.03 (−0.01, 0.06)	-
60–79	1.30 (0.87, 1.77) *	1.20 (0.76, 1.67) *	0.10 (0.04, 0.16) *	7.6 (3.2, 14.0) *
Sex				
Male	0.92 (0.49, 1.28) *	0.83 (0.41, 1.20) *	0.09 (0.04, 0.14) *	9.6 (4.5, 20.0) *
Female	0.90 (0.50, 1.32) *	0.88 (0.48, 1.29) *	0.03 (−0.01, 0.06)	-

Model adjusted for age, sex, race, education, income, total energy intake, alcohol intake, smoking, physical activity, BMI, self-reported hypertension, self-reported diabetes, and self-reported cardiovascular disease; * *p* < 0.05.

## Data Availability

The datasets utilized and scrutinized during the present inquiry are openly accessible in the NHANES repository.

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
