# Peer review of "Serum Anti-Aging Protein α-Klotho Mediates the Association between Diet Quality and Kidney Function"

_nutrients, 2023, doi:10.3390/nu15122744_

Round 1

Reviewer 1 Report

The study includes a high number of participants, which is an advantage.

More details about the mediation analysis should be provided.

How do the authors explain the association of 'added sugar' (incorrectly in table 2 as 'add sugar') with HEI2015 and eGFR?

Author Response

Point 1: The study includes a high number of participants, which is an advantage. More details about the mediation analysis should be provided.

Response 1: We thank the reviewer for the positive remarks about our study. We provided more detailed information including a path diagram of the mediation analysis in the Method. The revised text now reads as follows (Please see lines 147-156, 170-183): We explored the potential mediating roles of the aging process in the association between diet quality and kidney function. The causal mediation analysis was used for continuous exposure, mediator, and outcome with linear regression models (Figure 2). A bootstrap method was used to assess the confidence intervals for pathway estimates. Specifically, the HEI-2015 score was defined as the exposure variable, serum anti-aging protein α-Klotho as the mediator, and eGFR as the outcome variable. The following effects were measured: (i) the effect of diet quality on the anti-aging protein (Figure 2, path a), (ii) the effect of the anti-aging protein on kidney function (Figure 2, path b), (iii) the total effect of diet quality on kidney function (Figure 2, path c), and (iv) the direct effect of diet quality on kidney function, with the anti-aging protein considered (Figure 2, path cʹ). The proportion mediated was calculated using the formula: (ßtotal effectdirect effect)/ ßtotal effect × 100

    Please see the Figure 1 at the attachment

Figure 2. Path diagram of the mediation analysis model among 12,817 participants aged 40-79 years in the NHANES 2007–2016. Diet quality (measured by HEI-2015) was defined as an exposure; kidney function (measured by eGFR) was defined as an outcome; and aging process (measured by α-Klotho) was defined as a mediator. Path a indicates the regression coefficient for the association of HEI-2015 with α-Klotho. Path b indicates the regression coefficient for the association of α-Klotho with eGFR. Path c indicates the simple total effect of HEI-2015 with eGFR, without the adjustment for α-Klotho. Path c' indicates the direct effect of HEI-2015 with eGFR when controlled for α-Klotho. Abbreviation: NHANES, National Health and Nutrition Examination Survey; HEI, healthy eating index; eGFR, estimated glomerular filtration rate.

Point 2: How do the authors explain the association of 'added sugar' (incorrectly in table 2 as 'add sugar') with HEI2015 and eGFR?

Response 2: We appreciate the valuable suggestions provided by the reviewer. We have made a modification to the Tables, replacing “Add sugar” with “Added sugar”.

Numerous studies have indicated a potential association between excessive sugar consumption, particularly from added sugars and sugary beverages, and the development or progression of kidney disease [1,2]. In our study, we identified a positive association between added sugar consumption and both eGFR and UACR. However, it is important to mention that the positive association between added sugar and eGFR doesn’t imply that sugar intake is beneficial for kidney health. Several plausible explanations may account for this finding. Firstly, consuming high amounts of sugar can elevate plasma glucose levels, which in turn can lead to an increase in insulin levels. Elevated insulin levels may cause afferent renal vasodilation, leading to glomerular hyperfiltration and subsequently increasing eGFR [3,4]. Notably, increased eGFR has also been reported in patients in the early stage of type 2 diabetes mellitus [5]. Secondly, the possibility of reverse causality cannot be ruled out. Individuals at high risk of type 2 diabetes or CKD may have been aware of their condition through screening by general practitioners or their family history of these diseases. Consequently, the awareness of being at a high risk for those diseases might have motivated participants to avoids product that are high in sugar. However, given the cross-sectional design, it is premature to assert this definitive causal relationship. Thirdly, potential confounding by unmeasured lifestyle factors may contribute to the observed positive association. Research on the direct relationship between sugar intake and eGFR is limited, and more studies are required to better understand this association and to determine the underlying mechanisms involved.

References:

  1. Johnson RJ, Segal MS, Sautin Y, et al. Potential role of sugar (fructose) in the epidemic of hypertension, obesity and the metabolic syndrome, diabetes, kidney disease, and cardiovascular disease. Am J Clin Nutr. 2007;86:899-906.
  2. Karalius VP, Shoham DA. Dietary sugar and artificial sweetener intake and chronic kidney disease: a review. Adv Chronic Kidney Dis. 2013;20(2):157-64.
  3. Hostetter, T. H., Olson, J. L., Rennke, H. G., Venkatachalam, M. A. & Brenner, B. M. Hyperfiltration in remnant nephrons: a potentially adverse response to renal ablation. Am J Physiol.1981;241, F85–F93.
  4. Helal I, Fick-Brosnahan GM, Reed-Gitomer B, Schrier RW. Glomerular hyperfiltration: definitions, mechanisms and clinical implications. Nat Rev 2012;8(5):293-300.
  5. Bank N. Mechanisms of diabetic hyperfiltration. Kidney Int.1991;40(4):792-807.

Reviewer 2 Report

Line 68-75 The goal of this study should be rewritten. The study is examining if there is a relationship between kidney function and the levels of α-klotho. It is too preliminary to examine whether α-Klotho maintains good kidney function.

 Line 76-88 Include the age range, gender breakdown and if race or age was part of exclusion.

Table 1 A large percent  36.1 % of the study is identified as race is “others.”  The authors should try to separate out another race. For example, Hispanic which have a high risk of renal impairment.

Table 1 The range of ages needs to be identified.

Table S2.  Explain why some of the samples were eliminated (12,817) compared to the comparison for other parameters  which was 12, 821?

Author Response

Point 1: Line 68-75 The goal of this study should be rewritten. The study is examining if there is a relationship between kidney function and the levels of α-klotho. It is too preliminary to examine whether α-Klotho maintains good kidney function.

Response 1: We thank the reviewer for the concern regarding the goal of this study. We acknowledge that, given the cross-sectional design, it is premature to assert definitive hypothesis and goals. Consequently, we have revised the aim of this study as “Nevertheless, the role of serum α-Klotho in the relationship between diet quality and kidney function has not been fully elucidated. This study aimed to investigate whether serum α-Klotho mediates the relationship between a healthy dietary pattern and kidney function in a large cohort of middle-to-older aged individuals  from the National Health and Nutrition Examination Survey (NHANES) study conducted between 2007 and 2016. ” (Please see lines 71-76)

Point 2:  Line 76-88 Include the age range, gender breakdown and if race or age was part of exclusion.

Response 2: Thank you for your suggestion. We have provided more detailed information about exclusion. The revised text now reads as follows: “The initial cohort included 50,588 participants under the age of 80 years from NHANES 2007–2016. The serum α-Klotho level was assessed in participants aged 40-79 years old as part of the NHANES study. Subsequently, individuals without data on serum α-Klotho (n = 36,824), diet quality (n = 804), or kidney function (n = 143) were sequentially excluded. The final analytical cohort consisted of 12,817 participants aged between 40 and 79 years old.” (Please see lines 89-94). Participants were not excluded based on the gender or race.

Point 3:  Table 1 A large percent  36.1 % of the study is identified as race is “others.”  The authors should try to separate out another race. For example, Hispanic which have a high risk of renal impairment.

Response 3: We appreciate the valuable suggestions provided by the reviewer. We have made revisions to the race categories, which now include non-Hispanic White, non-Hispanic Black, Hispanic, or others, representing 44.1%, 19.7%, 27.4%, and 8.7% respectively. Given the change in the race categories, we have conducted new analyses by adjusting for the updated race categories. Notably, upon adjusting for the revised race groups, the main findings remain consistent, with only minor variations observed in β coefficient (95%CI). To reflect these updates, we have made necessary adjustments to the race categories in the Method section and Table 1, as well as β coefficient (95%CI) throughout the manuscript and supplementary files.

Point 4: Table 1 The range of ages needs to be identified.

Response 4: We thank the review for the suggestions. We have made adjustments to the title of Tables. It now reads as follows: “Table 1. Characteristics of 12,817 participants aged 40-79 years in the NHANES from 2007 to 2016”.

Point 5: Table S2.  Explain why some of the samples were eliminated (12,817) compared to the comparison for other parameters  which was 12, 821?

Response 5: We sincerely apologize for the oversight. Upon reevaluating our datasets, we discovered an additional 3 missing values for kidney function from the 2011-2012 datasets, bringing the total number of missing values to 21 instead of the previously reported 17 in the flowchart. We would like to clarify that the total number of participants in this study is 12,817. The error has been rectified in the updated flowchart, tables, and text accordingly.

Reviewer 3 Report

The manuscript by Cai et al. is related to the study of alpha-klotho, which is known to be a protective protein for the kidney.

Here the authors analysed the involvement of alpha-klotho in the association between a healthy diet and kidney function.

The study is well-designed and described.

The English language is clear and the form is correct.

Author Response

We would like to express our gratitude to the reviewer for the positive feedback and acknowledgment of our manuscript. We appreciate the time and effort invested in reviewing our work.

Round 2

Reviewer 2 Report

Appreciate the modifications of the manuscript. This improved the quality of the study.